# Improvements in Treatment Adherence after Family Psychoeducation in Patients Affected by Psychosis: Preliminary Findings

**DOI:** 10.3390/jpm13101437

**Published:** 2023-09-26

**Authors:** Salvatore Iuso, Melania Severo, Nicoletta Trotta, Antonio Ventriglio, Pietro Fiore, Antonello Bellomo, Annamaria Petito

**Affiliations:** 1Department of Humanistic Studies, University of Foggia and Italy, 71122 Foggia, Italy; iuso.salvatore@libero.it; 2Department of Clinical and Experimental Medicine, Viale Pinto, Policlinico Riuniti di Foggia, University of Foggia, 71122 Foggia, Italy; melania.severo93@gmail.com (M.S.); n.trotta2@gmail.com (N.T.); antonello.bellomo@unifg.it (A.B.); annamaria.petito@unifg.it (A.P.); 3Department of Clinical and Surgical Sciences, University of Foggia, 71122 Foggia, Italy; pietro.fiore@unifg.it; 4Neurorehabilitation and Spinal Unit, Institute Clinic Scientific Maugeri IRCCS, 70124 Bari, Italy

**Keywords:** family psychoeducation, Falloon, Gestalt, adherence, psychosis

## Abstract

(1) Background: Family psychoeducation is a well-recognized intervention which aims to improve the outcomes of illness in patients affected by psychosis. It has benefits in treatment adherence and leads to a reduction in relapses, higher levels of patient insight, and lower levels of stress within the family and among caregivers. (2) Methods: Eight patients and their families were recruited and randomly assigned to a Falloon-based family psychoeducation (FPP) intervention, and nine patients and their families were randomized to a Gestalt-based family intervention (GT). We compared the outcomes of these two treatment groups at a baseline assessment (T0), at the end of the programs (T1), and 6 and 12 months after the end of the programs (T2 and T3). The assessments included examinations of cognition (The Mini-Mental State Examination (MMSE) and The Five Point Test (5 Point)), the psychopathology and severity of illness (The Brief Psychiatric Rating Scale (BPRS), The Positive and Negative Syndrome Scale (PANSS), and The Clinical Global Impression Scale (CGI)), expressed emotion in families (Expressed Emotionality (Family Questionnaire-EE)), patient quality of life (The World Health Organization Quality of Life—BREF (WHOQOL-B)), social functioning (The Personal Social Performance (SPS)), aggression (Modified Overt Aggression Scale (MOAS)), and treatment adherence (The Brief Medication Adherence Report Scale (BMARS)). The primary aim was to test whether the FFP vs. GT program was more effective in improving treatment adherence over time. (3) Results: treatment adherence improved much more in the FFP group over time at any follow-up: +43.1% at T1, +24.0% at T2, and +41.6% at T3. Other characteristics, including psychopathology and the clinical stability of the subject, did not change over time. (4) Discussion: Family psychoeducation based on the Falloon program was effective at improving treatment adherence and contributed to avoiding relapses in the long term. Further studies on larger samples should be conducted to confirm this evidence, and similar psychoeducational programs should be routinely promoted in the clinical setting.

## 1. Introduction

Psychotic disorders are often associated with significant impairments in personal functioning, family functioning, as well as in social, educational, and working areas [1]. In addition, stigma and discrimination against people affected by psychotic disorders are common and add further distress in every-day life [2]. It has been argued that the five-year outcome of illness after the onset of psychosis is characterized by serious difficulties in social integration, an 80% risk of relapse, and a 10% chance of suicide [3]. It is also widely described that a sudden discontinuation in treatments and poor adherence to medications are associated with higher rates of relapses, increased hospitalizations, and a higher number of suicide attempts [4,5,6,7]. Conti et al. [8] reported that among 11,797 patients affected by chronic psychosis, 72.1% discontinued medication for at least 30 days at least once per year. All these factors inevitably place a severe burden on patients’ families and relatives, and above all, on caregivers [9].

Substantial benefits in terms of illness outcomes and treatment adherence have been reported in studies based on interventions combining pharmacotherapy with psychosocial treatments [10,11,12]. In particular, Fallon et al. argued that family-based interventions involving family educational strategies are efficaciously employed to reduce the impact of environmental stressors [11]. One of the most reliable and structured models for the management of psychosis has been proposed and validated by Ian Falloon; it includes an integrated psychoeducational intervention based on cognitive behavioral therapy [13]. This intervention is based on the active involvement of a patient’s family members in the rehabilitation process, with the aim of promoting clinical and social recovery. Intervention strategies include: (a) caregiver-based stress management, improving problem-solving skills and the social support system, and facilitating the achievement of personal life goals; and (b) caregiver education regarding the management of stress and behaviors associated with positive and negative symptoms [14]. An extensive scientific literature search suggested that family interventions, such as family psychoeducation, are effective in preventing relapses in schizophrenia and bipolar disorder [14,15,16,17]. Specifically, family interventions for people with severe psychiatric disorders may efficaciously reduce the rates of clinical relapses by increasing adherence to pharmacotherapy and making improvements in patients’ experiences regarding care after psychoeducation [12]. A large trial involving 340 patients affected by psychosis in the mental health centers of the Italian National Health Service, entitled “Study on psychoeducation enhancing results of adherence in patients with schizophrenia (SPERA-S)”, was conducted in 2013 with an 18-month follow-up [18]. The intervention reduced the prevalence of non-adherence in the treated group, with an effect size of 0.45 SD. Previously, Veltro et al. [19] reported the findings of an Italian randomized controlled trial on a family intervention with 1-year and 11-year follow-ups, confirming a significant reduction in positive symptoms, hospitalizations, and improvements in self-care and autonomy in daily life. More recently, in 2023, Roncone et al. [20] demonstrated the relevance of interventions on the families and caregivers of patients affected by schizophrenia. In particular, their investigation reported positive effects in family functioning and the personal growth of caregivers.

The aim of this study was to further investigate the clinical effectiveness of a psychoeducational program based on Falloon’s model (Falloon’s psychoeducational program (FPP)), involving the families of patients affected by psychotic spectrum disorders, compared with a generic treatment (Gestalt-oriented family treatment (GT)) delivered to a control group of matched patients. Specifically, we tested if FPP was more effective than a GT intervention in terms of improving adherence to pharmacotherapy and reducing the risk of relapses and hospitalizations within a follow-up period of six and twelve months after the end of these programs.

## 2. Materials and Methods

### 2.1. Study Design

In this study, families of patients affected by psychosis in a stable phase of illness were recruited. A group of families were randomly assigned to FPP and compared with a control group, who were randomized to a GT family treatment. Thus, randomized, blinded enrollment was conducted, since the evaluators were not informed about the treatment group (FFP or GT) when the patients and families were assessed. Patients and families were consecutively enrolled at the Psychiatric Outpatients Unit of the University of Foggia from April to September 2019. Those families providing consent were then randomly assigned to one of these intervention groups. Nonetheless, due to the small sample size, this study may be not considered as a proper randomized clinical trial, but rather, a preliminary examination. The diagnosis of psychosis in this study, including psychotic disorders such as schizophrenia and psychoses not otherwise specified, was based on the Diagnostic and Statistical Manual of Mental Disorders (DSM 5-criteria), 5th ed.-Text Revision) [21] and confirmed by two expert clinicians (A.V. and A.B.) through the Mini International Neuropsychiatric Interview (MINI) [22]. We recruited 17 patients who agreed to join the protocol with their families. Eight families were randomly assigned to the FPP and nine families were randomized to the GT. Exclusion criteria included patients affected by mental retardation or any other severe cognitive impairment, patients affected by psychosis due to substance abuse or any medical condition, diagnosis of affective psychosis, co-morbidity with substance dependence, incomprehension of the Italian language among patients and family members, and inability or unwillingness to provide informed consent among patients or family members. The timeline of the study included the enrollment of patients and their families (T0; the enrollment period took three months, during which 17 patients/families were identified), the administration of the FPP and GT treatments for six months (T1 represents the end of treatments after six months), and follow-ups performed at six (T2) and twelve months (T3) after the end (T1) of the treatments. Thus, the benefits of FPP and GT were measured at T1 as well as T2 and T3. Patients and families were informed of their treatment group, whereas evaluators were blinded to treatments. 

### 2.2. Aim of the Study

The main aim of this preliminary examination was to test the effectiveness of Falloon-based family psychoeducation (FPP) compared to a control program based on the Gestalt-based approach (GT) in terms of improving treatment adherence among patients affected by psychosis. Our secondary aims included the measure of the impact of these interventions on patients’ psychopathology, cognition, quality of life, social functioning, family expressed emotion, and aggression over time.

### 2.3. Interventions

The psychoeducational sessions were performed in the Psychiatric Outpatients Unit of the University of Foggia by ad hoc trained psychologists and psychiatrists (including the authors). They employed the psychoeducational modules proposed by Ian Fallon in his manual published in 1992 [23]. Falloon’s Psychoeducational Program (FPP) is a standardized intervention structured in three parts: (a) diagnostic sessions; (b) informative sessions; and (c) educational sessions with communication training and the development of problem-solving skills [13]. In addition, Falloon’s models include the provision of information to families about their relative’s mental disorder, how to manage symptoms and the first signs of crisis, how to increase families’ ability to cope with stress, and mutual solidarity among family members [13]. Treatment sessions were conducted weekly for six months (1.5 h per session) [24].

The GT is a family group treatment based on the Gestalt approach, providing general information on psychosis and conducted with the same frequency as FPP [25]. GT treatment sessions were also conducted weekly for six months (1.5 h per session) and were divided in four phases: (a) a short and informative phase on selected topics concerning the main problems related to mental disorders; (b) family members were invited to discuss how they dealt with the described problem, whether their solution was effective and why; (c) family members were invited to imagine alternative solutions to the problem; and (d) a ten-minute conclusion in order to summarize the topics discussed during the meeting [25].

In general, family psychoeducation aims to reduce the impact of environmental stress on vulnerable individuals by improving their communication skills within the family, increasing their coping strategies, and enhancing their problem solving skills [26]. Psychoeducational modules for families generally provide: (a) information about the illness and symptoms; (b) coping strategies for crisis management; (c) knowledge about the implementation of a more comfortable environment in which support is provided by peers and professionals to family members [26]. The non-specific effects of family psychoeducation are the emotional support, empathic listening, and the implementation of therapeutic optimism [13,26]. In FPP, the specific effects include the improvement of the patient’s problem-solving, coping, and social skills and an increase in stress tolerance [26]. In GT, the specific effects essentially include the reduction of shame and guilt, achieved by explicitly addressing situational problems related to the disorder, and the modeling of therapist’s non-judgmental, empathic, and supportive behavior [26]. 

We considered each family to have adhered to the psychoeducational program when at least one family member participated in 70% of the meetings (12 out of 18 planned meetings).

### 2.4. Assessment

The primary aim of this study was to test the improvement of patients’ adherence to treatment, which was measured with a self-reporting scale questionnaire and a specific four-question interview (described below). All participants underwent clinical and functional evaluations in order to explore the primary and secondary outcomes of these treatments. The assessments were performed at intake (T0), at the end of treatments (T1), and six (T1) and twelve (T2) months after the end of both treatments. 

The assessments included:-The Mini-Mental State Examination by Folstein 1975 [27], or MMSE, is a simple pen-and-paper test of cognitive functioning; it explores patient’s orientation, concentration, attention, verbal memory, and naming and visuospatial skills. A total score ranging from 24 to 30 points indicates normal cognitive functioning; scores ranging between 18 and 23 indicate a mild/moderate cognitive impairment; and scores ≤ 17 indicate a severe cognitive impairment.-The Brief Medication Adherence Report Scale (BMARS), a shorter form of the MARS-10, was employed as a measure of treatment adherence in the clinical setting [28]. It includes five-items based on a yes/no self-reporting scoring system; total scores may vary from 0 (low medication adherence) to 5 (high medication adherence) [29].-The Personal Social Performance (PSP) scale assesses functioning across four dimensions (socially useful activities, personal and social relationships, self-care, and disturbing/aggressive behaviors), with instructions on how to assess the patient and assign a score. The score ranges from 1 to 100, with 100 indicating the highest level of patient functioning [30].-The World Health Organization Quality of Life—BREF (WHOQOL-Brief) is a 26-item tool used to measure patients’ quality of life. Each item is scored from 1 to 5. Higher scores indicate a better quality of life. This tool explores four domains of quality of life: physical health, psychological well-being, social relationships, and environment [31]. Physical health includes items on mobility, daily activities, functional capacity, energy, pain, and sleep. Psychological measures include self-image, negative thoughts, positive attitudes, self-esteem, mindset, learning ability, memory concentration, religion, and mental state. The domain regarding the social relations contains questions about personal relationships, social support, and sex life. The environmental domain explores financial resources, safety, health and social services, the physical environment in which one lives, recreational activities, and the general environment (noise, air pollution, transportation, etc.) [32].-The Positive and Negative Syndrome Scale (PANSS) measures the severity of symptoms in schizophrenia. It is a 30-item scale exploring the positive and negative symptoms of illness and their relationship with the global psychopathology [33]. The PANSS includes three subscales: the Positive Scale, the Negative Scale, and the General Psychopathology Scale. Each subscale is rated from 1 to 7 points, i.e., from absent to extremely severe. The score of each subscale is the sum of the responses, while the total PANSS score is the sum of the subscales [34]. A composite scale, as considered in this study, was scored by subtracting the negative score from the positive one (PANSS composite scale = PANSS positive syndrome scale score – PANSS negative syndrome scale score). This yielded an index ranging from −42 to +42, reflecting the degree of predominance/balance of positive and negative symptoms [34].-The Brief Psychiatric Rating Scale (BPRS) [35] is the most widely used scale to measure general psychopathology in patients affected by psychiatric conditions. The scale consists of 24 items to be rated on a seven-point severity scale ranging from “not present” to “extremely severe”. It is based on a clinical interview and the patient’s behavior. The patient’s family can also provide a behavioral report on the patient [36]. The BPRS measures psychiatric symptoms of depression, anxiety, and psychosis in both clinical and research settings [37].-The Clinical Global Impression Scale (CGI) assesses the severity of illness and its changes from baseline as consequence of treatments. The CGI severity assessment is provided on a seven-point scale; additionally, improvements are rated on a seven-point scale, where responses may range from “much improved” to “much worsened” [38].-The Five Point Test (5TT) is a neuropsychological test assessing figural fluency. The participant is asked to generate as many unique drawings as possible within a certain time limit [39]. The task to be performed is to produce as many different patterns as possible by connecting the dots in each square with one or more straight lines within two minutes. The correction is done by calculating some indices: the number of total drawings; the number of errors made; the number of unique drawings (UD); the number of strategies used, such as rotation (CS); and the error index (ErrI) as the proportion of the cumulative number of failed drawings to the number of total drawings [40].-The Family Questionnaire (FQ) is a 20-item, self-administered questionnaire that measures the Emotional state Expressed (EE) through two subscales: criticism and the excessive emotional involvement of family members toward patients with a mental illness [37]. Each item is rated on a four-point scale (1 = never/very rarely; 4 = very often). The FQ is scored by summing each item rating; higher scores on one or both subscales (criticality ≥ 24; emotional hyper involvement ≥ 28) indicate a high degree of expressed emotion [41].-The Modified Overt Aggression Scale (MOAS) assesses the presence of four types of aggressive behavior: verbal aggression, aggression against property, self-aggression, and physical aggression. Aggressive acts are scored on the basis of their severity, from 0 to 4. The value of each item is multiplied by a specific factor assigned to each category: 1 for verbal aggression, 2 for aggression against objects, 3 for aggression against self, and 4 for aggression against others. The total score ranges from 0 to 40, where a higher score indicates a greater presence of aggressive behaviors [42].

In order to support the use of each tool in a clinical setting in Italy, here, we summarize normative or validation studies previously conducted in Italy: The Mini-Mental State Examination (MMSE) by Magni et al. [43], The Five Point Test (5 Point) by Cattelani et al. [40], The Brief Psychiatric Rating Scale (BPRS) by Roncone et al. [36], Family Questionnaire-EE (Expressed Emotionality) by Ponti et al. [44], The Positive and Negative Syndrome Scale (PANSS) by Pancheri and Brugnoli [45], The World Health Organization Quality of Life—BREF (WHOQOL-B) by De Girolamo et al. [46], Personal Social Performance (SPS) by Morosini et al. [47], Modified Overt Aggression Scale (MOAS) by Margari [42], The Clinical Global Impression Scale (CGI) as internationally translated and validated [38], and The Brief Medication Adherence Report Scale (BMARS) by the SOLE study group [48].

### 2.5. Ethical Considerations

This study was approved by the Ethical Committee of Province of Foggia (protocol number: NP3173; approved with deliberation number 199/15 April 2019). All participants provided written informed consent after receiving detailed instructions regarding the study design, aims, and outcome evaluation. Participants did not receive any compensation and joined the research study voluntarily. The study protocol complied with the provisions of the Declaration of Helsinki of 1995 and its subsequent revisions.

### 2.6. Statistical Analysis

A statistical analysis was performed employing the Grand Prism 5 (San Diego, CA, USA) software. Means and standard deviations (SD) were calculated for each parameter. In order to detect any overall differences between the related means of the groups at different times (T0, T1, T2, T3), we employed ANOVA for repeated measures. Post hoc analysis was performed using the Bonferroni test. An alpha level of ≤0.05 was considered statistically significant throughout the study.

## 3. Results

Seventeen families of patients suffering from psychotic spectrum disorders were recruited and randomly assigned to either the experimental group (n = 8), based on Falloon’s protocol for psychoeducation with caregivers, or the control group, based on a generic Gestalt-based family treatment (n = 9). 

Patients recruited comprised 10 males (7 in the FPP and 3 in the GT control group) and 7 females (1 in the FPP and 6 in the GT-based intervention) with a mean age of 37.8 ± 9.37 years old (36.8 ± 8.70 years old in the FPP and 38.6 ± 10.8 years old in the GT-group). Males were aged 35.7 ± 7.71 years whereas females were 41.5 ± 12.1 years old. All patients depended upon their family of origin, as noted in the protocol, since they were not married or engaged in any relationship. Differences in current age, years of education, and rates of employment were not statistically significant at baseline (Table 1). Eight patients were on psychopharmacological monotherapy (two patients on a stable treatment with Olanzapine, two with Amisulpride, one with Risperidone, one with Paliperidone, one with Aripiprazole, and one with Clozapine), nine patients were on polypharmacotherapy (five patients treated with Clozapine and mood stabilizers, two with Risperidone and mood stabilizers, one with Quetipiane and Valproic Acid, and one with Amisupride and Clozapine).

The cognitive dimensions, assessed with the MMSE test and the 5-point test, did not show statistically significant differences between the two treatment groups at baseline (T0), at the end of treatment (T1), or in the two follow-up retests at six and twelve months (T2 and T3) (shown in Table 1). The mean MMSE scores indicated a moderate cognitive impairment among patients receiving the FFP intervention (23.75 ± 9.45), whereas patients in the GT-group reported severe cognitive impairment (17.67 ± 5.31). Nonetheless, the difference between the cognitive scores was not statistically significant (MMSE: *p* = 0.1389; 5-Point: *p* = 0.7361). Additionally, the distribution of subjects at baseline was randomly performed, so there was no selection-bias for cognitive impairment; follow-up showed that there was no influence of baseline cognition on the psychoeducational outcomes (Table 1).

Our assessment of psychopathology, based on BPRS and PANSS, described a significant level of general symptoms, as expected among patients affected by psychosis, albeit with a stable phase of illness (BPRS scores ranged from 47 to 51, with no significant differences between the two groups, *p* = 0.6399), with low levels of psychotic symptoms based on PANSS (13.3–13.7, *p* = 0.8794), confirming that all recruited patients were not acutely ill and their therapeutic trail was not significantly influenced by positive or negative symptoms. This evidence was confirmed by the CGI scores, reporting “mildly ill” as the general judgment of clinical severity. Finally, these scores (BPRS, PANSS, and CGI) did not differ significantly between the two groups at any follow-up after the interventions (Table 1).

There were no statistically significant differences among the treatment groups at baseline in terms of expressed emotions (FQ-EE: *p* = 0.1918), quality of life (WHOQOL-B: *p* = 0.1931), social functioning (SPS: *p* = 0.4290), aggression (MOAS: *p* = 0.3112), or treatment adherence (BMARS: *p* = 0.3749). These characteristics confirmed that both treatment groups (GT and FPP) were well matched at baseline, even if differences might be revealed with a larger sample. Additionally, the expressed emotions, quality of life, social functioning, and aggression assessments did not differ significantly at any follow-up after the psychoeducational interventions (GT and FPP) (Table 2). Both groups presented non-critical levels of expressed emotion (FQ-EE scores ranging from 15.0 to 18.9), low levels of aggression (MOAS ranging from 1.75 to 3.00), and medium levels of quality of life (WHOQOL-B ranging from 54.5 to 62.0) at baseline.

Regarding the primary outcome of this intervention, medication adherence, as scored by BMARS, was not different at baseline; this confirmed that all patients involved in the study showed a good initial rate of adherence to treatments. These levels of adherence differently improved at each retest and follow-up (T1, T2, T3, *p* < 0.0001), as confirmed in the post hoc analysis (Table 2). In particular, the analyses showed much higher levels of treatment adherence among the group undergoing Fallon’s psychoeducation-based intervention (T1; BMARS: 3.75 ± 0.46). The improvement achieved by the experimental group was maintained at the 6- (T2:3.25 ± 1.38) and 12-month follow-ups (T3:3.71 ± 0.48) (Table 2). According to this evidence, no patients discontinued their medications during this time, which may have contributed to the absence of clinical relapses.

## 4. Discussion 

Psychoeducation is a well-recognized intervention with many benefits in a psychiatric setting. It has been demonstrated that psychoeducation improves the outcome of illness in schizophrenia in terms of treatment adherence, with a consequent reduction of clinical relapses and higher levels of patient insight [49,50]. Additionally, psychoeducation has been demonstrated to be effective in reducing aggression and improving anger expression in various setting, as well as in managing illness-related stress, leading to better psychosocial functioning [51,52]. Family interventions, including psychoeducation and family behavioral therapy, reduce the emotional burden of relatives and family members [53]. In particular, behavioral family management based on Falloon’s model was shown to be significantly more effective at reducing caregivers’ expressed emotion than the other interventions [20]. Nonetheless, recent studies have also reported different results regarding the effectiveness of family psychoeducation in Japanese families of patients affected by schizophrenia, with no significant reduction of emotionality being observed among caregivers [54].

Our sample was composed of eight patients/families randomly assigned to a Falloon-based protocol of psychoeducation vs. nine patients/families undergoing a generic Gestalt-based family treatment. Both groups of patients were randomly assigned and did not show statistically significant differences in terms of baseline cognition, psychopathology, expressed emotion, aggression, social functioning, quality of life, or treatment adherence (Table 1). It is worth noting that this is methodologically controversial, since the measurements of outcome were not apparently influenced by baseline characteristics, because of the small sample size. Similar levels of baseline cognition and psychopathology confirmed that all patients were equally prone to receive benefits from family psychoeducation. In addition, we did not find any significant differences at baseline or at the follow-ups in the levels of cognition and psychopathology. We may argue that neither intervention led to an improvement in clinical conditions whilst also promoting a stable illness course with no relapses over a long follow-up period (as confirmed by the scores of BPRS, PANSS, and CGI up to T3 follow-up). This may be considered a relevant finding, since the course of psychosis is characterized by a high risk of relapses [49].

Patients in both treatment groups reported similar baseline levels of social functioning (SPS scores), quality of life (WHOQOL-B scores), aggression (MOAS), and expressed emotion (FQ-EE). This is of interest in terms of our comparison between the two groups. We did not find any significant change in any of these characteristics in either group over time or between the groups. This finding may reflect the stability of the sampled patients and may indicate the absence of additional benefits of Falloon-based psychoeducation in these specific domains [54].

In contrast, we found a statistically significant impact of Falloon-based psychoeducation on the treatment adherence of patients/families (BMARS scores). Although there was no difference in the levels of adherence at baseline between the two groups (1.88 ± 1.61 vs. 2.62 ± 1.68; *p* = 0.3749), they improved much more in the FFP group after the intervention at any follow-up: +43.1% at T1 (ΔT1−T0), +24.0% at T2 (ΔT2−T0), and +41.6% at T3(ΔT3−T0) (Table 2). Improvements in adherence among patients receiving GT intervention were as follows: +6.3% at T1(ΔT1−T0), −29.2 at T2 (reduction of adherence as ΔT2−T0), and +6.3% at T3 (ΔT3−T0). These findings clearly show a dubious effectiveness of GT intervention in terms of improving treatment adherence, with a partial reduction at T2, but also confirmed the significant superiority of FPP in terms of improving adherence, with a stable result over time within 12 months after the end of the program (increased adherence of 41.6%; *p* < 0.0001; Table 2).

Our findings may suggest the effectiveness of Falloon-based family psychoeducation (FPP) in terms of improving treatment adherence among clinically stable patients affected by psychosis [10,11,12,13]. This improvement was stable over time (T0–T3), but this evidence is limited by the small sample size and the absence of a non-psychoeducation comparison group. Nonetheless, FFP had a significantly higher impact on adherence than GT. The observed improvements in treatment adherence after psychoeducation based on Falloon’s methods might be attributable to the following factors: an increase in patient health/illness awareness and insight; higher competence in managing stressful events and expressed emotion on a patient and family level, with a protective role of psycho-pharmacotherapy; higher knowledge about the characteristics of psychosis and the role of psychotropics; higher competence of the family in terms of monitoring patient treatment adherence; and higher levels of patient motivation for self-care [11].

The limitations of this study include the small number of subjects, even if family interventions are typically addressed to a limited number of patients (considering the group of relatives and caregivers involved). The small sample size may have affected the absence of baseline differences among the two groups; the lack of additional information regarding the history of illness and lifetime number of episodes, hospitalizations, relapses and so on; the lack of feedback from families about their own burden beyond expressed emotion; the lack of any questionnaire regarding insight or illness awareness among patients and families; and the lack of a third control group with no family psychoeducation.

The strong points of this study include its randomized sampling, even though it was conducted on a small sample; the comparison of FPP with a control group (GT); all patients and families completed all sessions as planned in both programs; all patients completed the assessment at baseline and at all follow-ups; there were no drop-outs over the study period; and long-term follow-up and assessment, i.e., 1 year after the end of program.

## 5. Conclusions

We conclude that family psychoeducation based on the Falloon program is effective at improving treatment adherence among patients affected by psychosis and that it, as well as the GT-program, possibly contributed to preventing clinical relapses in the long-term and supporting patients’ caregivers in the management of illness. Further studies on larger samples, including a non-psychoeducation comparison group, should be conducted in order to confirm this preliminary evidence, and similar psychoeducational programs should be routinely promoted in the clinical setting, above all for families of patients affected by severe mental illness.

## Figures and Tables

**Table 1 jpm-13-01437-t001:** Characteristics at baseline (T0) and differences between the Falloon’s Psychoeducational Program (FPP) group and control (GT, Gestalt-oriented family treatment) group.

Characteristics at T0, Baseline		*p*-Value
*GT*	*FPP*	
** * Patients’ current age * **		
38.6 ± 10.8	36.8 ± 8.70	0.7074
** *Sex (males)* **		
3 (30%)	7 (70%)	0.0235
** *Education (years)* **		
11.4 ± 3.67	12.3 ± 4.86	0.6606
** *Employment (yes)* **		
4 (66.7%)	2 (33.3%)	0.4024
** *The Mini-Mental State Examination (MMSE)* **		
17.6 ± 5.31	23.7 ± 9.45	0.1389
** *5 Point (The Five Point Test)* **		
4.77 ± 2.86	5.12 ± 0.83	0.7361
** *BPRS (The Brief Psychiatric Rating Scale* ** *)*		
47.0 ± 12.2	51.2 ± 22.1	0.6399
** *Family Questionnaire-EE (Expressed Emotionality)* **		
18.9 ± 3.75	15.0 ± 7.03	0.1918
** *PANSS (The Positive and Negative Syndrome Scale)-composite scale* **		
13.7 ± 5.01	13.3 ± 5.65	0.8794
** *WHOQOL-B (* ** ** *The World Health Organization Quality of Life—BREF)* **		
54.5 ± 7.90	62.0 ± 13.23	0.1931
** *SPS (The Personal Social Performance)* **		
42.4 ± 13.1	48.0 ± 14.7	0.4290
** *MOAS (Modified Overt Aggression Scale)* **		
3.00 ± 2.82	1.75 ± 2.05	0.3112
** *CGI (The Clinical Global Impression Scale)* **		
3.88 ± 1.16	3.62 ± 1.56	0.6955
** *BMARS (The Brief Medication Adherence Report Scale)* **		
1.88 ± 1.61	2.62 ± 1.68	0.3749

**Table 2 jpm-13-01437-t002:** Differences in characteristics between the control group (GT, Gestalt-oriented family treatment) and Falloon’s Psychoeducatinal Program (FPP) group at baseline (T0), after the intervention (T1), 6 months (T2), and 12 months from the end of the intervention (T3).

T0, *Baseline*		T1, *after the Intervention*		T2, *6 Months from the End*		T3, 12 *Months from the End*		** *F* **	***p*-Value**	** *Post Hoc Test* ** ** *(Bonferroni)* **
*GT*	*FPP*	*GT*	*FPP*	*GT*	*FPP*	*GT*	*FPP*			***p*-Value**
*The Mini-Mental State Examination (MMSE)*										
17.6 ± 5.31	23.7 ± 9.45	15.1 ± 4.91	23.6 ± 13.09	15.3 ± 4.06	19.6 ± 6.67	14.6 ± 3.46	20.0 ± 8.52	0.1821	0.9082	0.4376
** *5 Point (The Five Point Test)* **										
4.77 ± 2.86	5.12 ± 0.83	4.11 ± 2.31	4.25 ± 1.48	3.55 ± 1.74	4.00 ± 1.30	3.82 ± 1.64	3.28 ± 1.38	0.2535	0.8585	0.6753
** *BPRS (The Brief Psychiatric Rating Scale)* **										
47.0 ± 12.2	51.2 ± 22.1	39.8 ± 8.89	48.3 ± 27.4	48.5 ± 12.3	51.7 ± 22.1	49.1 ± 14.0	54.7 ± 24.7	0.0643	0.9785	0.2235
** *Family Questionnaire-EE (Expressed Emotionality)* **										
18.9 ± 3.75	15.0 ± 7.03	18.4 ± 6.48	18.1 ± 8.02	19.4 ± 7.00	19.5 ± 6.56	18.3 ± 6.36	22.0 ± 4.35	1.0240	0.3883	0.4563
** *PANSS (The Positive and Negative Syndrome Scale)-composite scale* **										
13.7 ± 5.01	13.3 ± 5.65	12.2 ± 3.83	13.8 ± 5.66	14.1 ± 4.62	14.7 ± 4.92	14.3 ± 4.03	15.8 ± 6.89	0.1408	0.9351	0.7658
** *WHOQOL-B (The World Health Organization Quality of Life—BREF)* **										
54.5 ± 7.90	62.0 ± 13.23	58.4 ± 11.4	58.3 ± 13.1	53.8 ± 11.6	52.3 ± 8.34	51.3 ± 11.58	49.1 ± 11.8	0.6657	0.5764	0.5757
** *SPS (The Personal Social Performance)* **										
42.4 ± 13.1	48.0 ± 14.7	48.2 ± 14.8	48.5 ± 15.4	46.8 ± 16.1	43.7 ± 14.0	43.3 ± 16.8	37.0 ± 11.3	0.5103	0.6767	0.5234
** *MOAS (Modified Overt Aggression Scale)* **										
3.00 ± 2.82	1.75 ± 2.05	3.22 ± 3.23	1.75 ± 2.25	4.33 ± 3.24	3.37 ± 3.37	5.25 ± 3.45	3.42 ± 2.63	0.0659	0.9777	0.2387
** *CGI (The Clinical Global Impression Scale)* **										
3.88 ± 1.16	3.62 ± 1.56	3.77 ± 1.09	3.75 ± 1.66	4.00 ± 0.70	4.00 ± 1.51	3.75 ± 1.03	4.00 ± 1.73	0.1042	0.9573	0.1435
** *BMARS (The Brief Medication Adherence Report Scale)* **										
1.88 ± 1.61	2.62 ± 1.68	2.00 ± 0.86	3.75 ± 0.46	1.33 ± 0.76	3.25 ± 1.38	2.00 ± 1.06	3.71 ± 0.48	0.9496	**<0.0001**	**<0.0001**

## Data Availability

Data sharing is not applicable to this study because of privacy or ethical restrictions.

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
