# Peer review of "Improvements in Treatment Adherence after Family Psychoeducation in Patients Affected by Psychosis: Preliminary Findings"

_jpm, 2023, doi:10.3390/jpm13101437_

Round 1

Reviewer 1 Report

The study aimed to describe the outcomes of “Improving the treatment adherence to pharmacotherapy, reducing the risk of relapses and rehospitalization”… at 1-year follow-up.

The work, not original, given that there have been numerous contributions in the field of psychoeducation in schizophrenia and psychosis, could have brought, if rigorous, a small contribution to improving the understanding of the interventions' effectiveness.

The work, to be understood as preliminary, presents very serious criticalities and cannot be accepted.

INTRODUCTION

The bibliographic references have not been carefully chosen concerning the topics of the work and need to be updated. The paragraph must be entirely improved considering the most recent international and Italian literature.

METHODOLOGY

The methodology (with randomization) was described (with poor accuracy) 2 times in the text.

Interventions

??? Manuals in Italian? Operator training in this regard?

Where did the sessions take place?

Assessment

The instruments used are numerous; in our opinion, some of these need to be validated in our country or are inappropriate for this study.

The tools listed are redundant in their number and not appropriate.

Both the MMSE and the Five Point Test (both used in clinical neuropsychology for l) do not appear to be suitable tools in the context of a protocol for the evaluation of a family psychoeducational intervention in a sample of young subjects (average age 35-40) affected by psychosis.

Similarly, it is suggested not to report the MOAS on aggression (unjustified and stigmatizing unless its inclusion is justified).

The validation in Italian is only reported for some instruments, such as for the Family Questionnaire.

No assessment tools are reported for the family members involved in the interventions.

Statistical analysis

The Authors should have used general linear models for repeated measures analyses with a between-subjects factor (G1, G2) and a within-subjects factor (pre-treatment vs. post-treatment) for considered outcome variables.

DISCUSSION

It seems there was a mistake in loading the work text of the work…. in this section, results and Table 1-2 were repeated.

The findings should have been discussed, comparing their results with the literature findings.

Minor editing of English language required

Author Response

#3 reviewer

Changes in blue in the text

 The study aimed to describe the outcomes of “Improving the treatment adherence to pharmacotherapy, reducing the risk of relapses and rehospitalization”… at 1-year follow-up.

The work, not original, given that there have been numerous contributions in the field of psychoeducation in schizophrenia and psychosis, could have brought, if rigorous, a small contribution to improving the understanding of the interventions' effectiveness.

The work, to be understood as preliminary, presents very serious criticalities and cannot be accepted.

R: Thank you. We have now presented these findings as preliminary in the title and the text. We agree that the study may be not so original but comparisons made between FFP and another active intervention (GT) was not very common in the international literature. Also the study design was set a 1-year follow-up and this may add strength.

INTRODUCTION

The bibliographic references have not been carefully chosen concerning the topics of the work and need to be updated. The paragraph must be entirely improved considering the most recent international and Italian literature.

R: thank you. the introduction deals, in order, with these topics: relevance of treatments and treatment adherence in the course of psychosis; Falloon’s models on family intervention; a new section of some Italian experiences has been added. Most of studies by Falloon have been conducted in the 80-90s. Also, other studies on adherence and family intervention in Italy (e.g. S-PERAS trial) are not recent. FFP and treatment adherence is not so specifically explored  in the previous literature

METHODOLOGY

The methodology (with randomization) was described (with poor accuracy) 2 times in the text.

R: more details have been added in the study design

Interventions

??? Manuals in Italian? Operator training in this regard?

Where did the sessions take place?

R: all details have been added about manual, training and place

Assessment

The instruments used are numerous; in our opinion, some of these need to be validated in our country or are inappropriate for this study.

R: all references about normative and validation studies in Italian clinical samples are reported in a specific paragraph for each tool

The tools listed are redundant in their number and not appropriate.

R: we believe that clinical, cognitive assessment are always appropriate in severely mentally ill patients followed over time. Specific tools on social performance, aggression, EE, quality of life and adherence are all useful instruments in order to explore all areas of functioning of patients and their family (family questionnaire)

Both the MMSE and the Five Point Test (both used in clinical neuropsychology for l) do not appear to be suitable tools in the context of a protocol for the evaluation of a family psychoeducational intervention in a sample of young subjects (average age 35-40) affected by psychosis.

R: cognitive assessment at baseline is useful to test if patients are prone to receive benefits from psychoeducation and exclude those reporting a cognitive impairment due to the illness. A follow-up of cognition is also useful since cognitive symptoms are core in psychoses.

Similarly, it is suggested not to report the MOAS on aggression (unjustified and stigmatizing unless its inclusion is justified).

R: patients’ aggression is an important outcome measure for family interventions since it directly reflects EE and family functioning/stress.

The validation in Italian is only reported for some instruments, such as for the Family Questionnaire.

R: all references about normative and validation studies in Italian clinical samples are reported in a specific paragraph for each tool

No assessment tools are reported for the family members involved in the interventions.

R: Family Questionnaire is a report from family. Other family-tools were not employed and this was stated in the limitations. The primary aim of the investigation was to test patients’ adherence

Statistical analysis

The Authors should have used general linear models for repeated measures analyses with a between-subjects factor (G1, G2) and a within-subjects factor (pre-treatment vs. post-treatment) for considered outcome variables.

R: We repeated the analyses employing ANOVA for repeated measures  and reported new statistical parameters (in yellow). Significant differences previously detected across the time were confirmed and values were changed in the text and table. Further levels of analyses were not performed since the sample size was small

DISCUSSION

It seems there was a mistake in loading the work text of the work…. in this section, results and Table 1-2 were repeated.  

R: results and tables were re-cited for a clearer discussion

The findings should have been discussed, comparing their results with the literature findings.

R: findings have been compared with references 49-54.

Reviewer 2 Report

This manuscript entitled “Improvements in Treatment Adherence after a Family Psychoeducation Intervention in Patients affected by Psychosis” deals with a clinical trial that provides family intervention in patients with psychosis. However, there are overall concerns about what is presented in this manuscript because there are several serious concerns about the methods and results.

This study would be more of a preliminary examination than a randomized clinical trial. It is difficult to indicate the effectiveness of the targeted family intervention in this study. The authors described both groups as randomized and matched for baseline data, but serious differences exist. This seems to be due to the small sample size, which did not show significant differences. Furthermore, the method of analysis needed to be more appropriate. The Kruskal-Wallis test is not appropriate for repeated measures data. Some of the data in the results seem questionable. For example, the PANSS scores do not seem appropriate; since there are 30 items, the minimum score should be 30.

The psychosocial interventions proposed by Professor Ian Falloon are a revolution in schizophrenia treatment, and it is fascinating that efforts are being made to establish their effectiveness. Calculating sample size by power analysis and conducting clinical trials using appropriate procedures would be desirable. The only thing that can be said about the present sample is that it is feasible as a preliminary examination.

Author Response

#1 reviewer

Changes in yellow in the text

This manuscript entitled “Improvements in Treatment Adherence after a Family Psychoeducation Intervention in Patients affected by Psychosis” deals with a clinical trial that provides family intervention in patients with psychosis. However, there are overall concerns about what is presented in this manuscript because there are several serious concerns about the methods and results.

This study would be more of a preliminary examination than a randomized clinical trial.

R: this is correct and has been specified within the text of the MS. Even if randomization has been employed, the sample size does not reach the evidence of  RCT. Thank you.

It is difficult to indicate the effectiveness of the targeted family intervention in this study. The authors described both groups as randomized and matched for baseline data, but serious differences exist. This seems to be due to the small sample size, which did not show significant differences.

R: this is a very good point since it lets us implement the discussion of our findings. Since the sample size is not modifiable and comparisons made at baseline did not show statistically significant differences, we added these comments in the results/ discussion and limitations sections. Thank you

 Furthermore, the method of analysis needed to be more appropriate. The Kruskal-Wallis test is not appropriate for repeated measures data.

R: thanks. We repeated the analyses employing ANOVA for repeated measures  and reported new statistical parameters. Significant differences previously detected across the time were confirmed and values were changed in the text and table.

Some of the data in the results seem questionable. For example, the PANSS scores do not seem appropriate; since there are 30 items, the minimum score should be 30.

R: this is a very good point and we missed the correct description. In fact, we employed the PANSS Composite Scale which is scored by subtracting the negative score from the positive score (PANSS composite scale = PANSS positive syndrome scale score—PANSS negative syndrome scale score). This yields an index ranging from –42 to +42, reflecting the degree of predominance/balance of positive and negative symptoms. We added this description in the text and added specifiers in the tables

The psychosocial interventions proposed by Professor Ian Falloon are a revolution in schizophrenia treatment, and it is fascinating that efforts are being made to establish their effectiveness. Calculating sample size by power analysis and conducting clinical trials using appropriate procedures would be desirable.

R: this is a rewarding comment. We believe that Falloon’s interventions should be promoted in the routine clinical setting. We called for further studies on larger samples in the conclusions

The only thing that can be said about the present sample is that it is feasible as a preliminary examination.

R: we specified that this was a preliminary examination in the title and within the whole manuscript

Reviewer 3 Report

Comments to authors:

Abstract 

Please add the aim of this study in abstract.

Introduction

Well written

Method

Is there more specific diagnosis of psychotic spectrum disorders (ex, schizophrenia, schizoaffective disorder, and delusional disorder etc)? Authors need to insert more detail demographic and clinical data not only patients but also caregivers before Table 1 if they have data. (Ex, age, sex, disease period, working or not, and relationship of patients). In any case, authors should need to add a table summarizing further demographic data.

Discussion

Authors claim that GT and FPP contributed to a stable 12 months outcome, however it is not reasonable. That is because in this study, authors did not set up non-psychoeducation group. Therefore, although it is extremely likely that psychoeducation is effective in stabilizing from prior study, the above interpretation cannot be made as a conclusion of only this study. 

The main finding in this study is that Falloon-based psychoeducation is more effective than GT in terms of adherence. Authors should describe the reasonable explain what factor make such result.

Others

The authors' method of analysis is not bad; however, it is poor. Often, repeated measures ANOVA is used in time series analysis. In recently, it is recommended to use analysis methods such as mixed-effects models for more suitable method. 

This study has a strength in terms of long-term follow up data even though sample number is a few. It is regrettable that more interesting results could have been obtained by analyzing each item by time. I recommend more suitable analysis be used the next time. As a reference for analysis of time series data, please refer to the following paper. (Title: Placebo-Controlled Trial of Amantadine for Severe Traumatic Brain Injury)

Author Response

#2 reviewer

Changes in green in the text

Abstract 

Please add the aim of this study in abstract.

R: the primary aim of the study has been stated and highlighted in green in the abstract; the secondary ones were not reported but included in the full text in order to meet the number of words allowed for the abstract

Introduction

Well written

R: thanks a lot

Method

Is there more specific diagnosis of psychotic spectrum disorders (ex, schizophrenia, schizoaffective disorder, and delusional disorder etc)?

R: diagnoses were better specified in the text (schizophrenia and psychoses NOS) and version of DSM (5) referenced correctly

Authors need to insert more detail demographic and clinical data not only patients but also caregivers before Table 1 if they have data. (Ex, age, sex, disease period, working or not, and relationship of patients). In any case, authors should need to add a table summarizing further demographic data.

R: many thanks; additional sociodemographics (education, employment, relationships…..) were commented in the text and added in table 1 with more statistical notes.

Discussion

Authors claim that GT and FPP contributed to a stable 12 months outcome, however it is not reasonable.

R: this is a relevant point. All these limitations have been added in the discussion and conclusions

That is because in this study, authors did not set up non-psychoeducation group. Therefore, although it is extremely likely that psychoeducation is effective in stabilizing from prior study, the above interpretation cannot be made as a conclusion of only this study. 

R: this is a relevant point. All these limitations have been added in the discussion and conclusions

The main finding in this study is that Falloon-based psychoeducation is more effective than GT in terms of adherence. Authors should describe the reasonable explain what factor make such result.

R: a paragraph on factors promoting higher treatment adherence was added in the discussion. Thank You

Others

The authors' method of analysis is not bad; however, it is poor. Often, repeated measures ANOVA is used in time series analysis. In recently, it is recommended to use analysis methods such as mixed-effects models for more suitable method. This study has a strength in terms of long-term follow up data even though sample number is a few. It is regrettable that more interesting results could have been obtained by analyzing each item by time. I recommend more suitable analysis be used the next time. As a reference for analysis of time series data, please refer to the following paper. (Title: Placebo-Controlled Trial of Amantadine for Severe Traumatic Brain Injury)

R: We repeated the analyses employing ANOVA for repeated measures  and reported new statistical parameters (in yellow). Significant differences previously detected across the time were confirmed and values were changed in the text and table. Further levels of analyses were not performed since the sample size was small

Round 2

Reviewer 2 Report

Thank you for the revision regarding the comments. The limitations of this study were carefully described. On the other hand, the following points need to be reconsidered.

It needs to be clarified what the results of the repeated measures ANOVA shown in Table 2 are. Do they indicate that the interactions were significant? The post hoc results should also be shown.

The authors replied that they revised the title to reflect that this was a preliminary study, but it has yet to be changed.

Author Response

Thank you for the revision regarding the comments. The limitations of this study were carefully described. On the other hand, the following points need to be reconsidered.

R: thanks for accepting our changes

It needs to be clarified what the results of the repeated measures ANOVA shown in Table 2 are. Do they indicate that the interactions were significant? The post hoc results should also be shown.

R: P-values indicate those interactions which were significant. The results have been implemented in the table 2 and post-hoc analyses based on Bonferroni reported.

The authors replied that they revised the title to reflect that this was a preliminary study, but it has yet to be changed.

R: the title has been changed into “Improvements in Treatment Adherence after  Family Psychoeducation in patients affected by Psychosis: preliminary findings”

Reviewer 3 Report

The authors have addressed my concerns  adequately.

The authors state that they did not perform further analysis due to the small sample size, but this is a lack of understanding. Note that the mixed-effects model is more useful than repeated ANOVA for small samples because it accounts for individual differences and is robust to missing values.

Author Response

The authors have addressed my concerns  adequately.

R: many thanks for accepting our changes

The authors state that they did not perform further analysis due to the small sample size, but this is a lack of understanding. Note that the mixed-effects model is more useful than repeated ANOVA for small samples because it accounts for individual differences and is robust to missing values.

R: we revised the paper employing the repeated measures ANOVA as suggested by more than one reviewer. This choice did not exclude the mixed-effects model which is a very helpful suggestion for the analysis of further findings from this trial 

Round 3

Reviewer 2 Report

Thank the authors for addressing all the comments.